# Detection of Equine Papillomaviruses and Gamma-Herpesviruses in Equine Squamous Cell Carcinoma

**DOI:** 10.3390/pathogens12020179

**Published:** 2023-01-23

**Authors:** Lea Miglinci, Paul Reicher, Barbara Nell, Michelle Koch, Christoph Jindra, Sabine Brandt

**Affiliations:** 1Research Group Oncology (RGO), Clinical Unit of Equine Surgery, Department for Companion Animals and Horses, Veterinary University, 1210 Vienna, Austria; 2Clinical Unit of Ophthalmology, Department for Companion Animals and Horses, Veterinary University, 1210 Vienna, Austria; 3Division of Molecular Oncology and Hematology, Karl Landsteiner University of Health Sciences, 3500 Krems an der Donau, Austria

**Keywords:** horse, squamous cell carcinoma, SCC, equine papillomavirus, EcPV2, herpesviruses

## Abstract

Squamous cell carcinoma (SCC) seriously compromises the health and welfare of affected horses. Although robust evidence points to equine papillomavirus type 2 (EcPV2) causing genital lesions, the etiopathogenesis of equine SCC is still poorly understood. We screened a series of SCCs from the head-and-neck (HN), (peri-)ocular and genital region, and site-matched controls for the presence of EcPV2-5 and herpesvirus DNA using type-specific EcPV PCR, and consensus nested herpesvirus PCR followed by sequencing. EcPV2 DNA was detected in 45.5% of HN lesions, 8.3% of (peri-)ocular SCCs, and 100% of genital tumors, whilst control samples from tumor-free horses except one tested EcPV-negative. Two HNSCCs harbored EcPV5, and an ocular lesion EcPV4 DNA. Herpesvirus DNA was detected in 63.6%, 66.6%, 47.2%, and 14.2% of horses with HN, ocular, penile, and vulvar SCCs, respectively, and mainly identified as equine herpesvirus 2 (EHV2), 5 (EHV5) or asinine herpesvirus 5 (AsHV5) DNA. In the tumor-free control group, 9.6% of oral secretions, 46.6% of ocular swabs, 47% of penile samples, and 14.2% of vaginal swabs scored positive for these herpesvirus types. This work further highlights the role of EcPV2 as an oncovirus and is the first to provide information on the prevalence of (gamma-)herpesviruses in equine SCCs.

## 1. Introduction

Genetic factors and exogenous triggers including UV radiation, toxic substances and pollutants are accepted drivers of cancer development and progression. In addition, viruses are increasingly recognized as cancer-promoting agents, including papillomaviruses (PVs) and herpesviruses [1,2]. PVs are a family of small non-enveloped dsDNA viruses. They are usually species-specific and have a pronounced tropism for keratinocytes. PVs can induce benign epithelial lesions or malignant cancers in humans and a wide range of animals, with PV oncoproteins E6, E7, and frequently E5 driving neoplastic transformation [3]. Herpesviruses are highly complex enveloped viruses containing a large linear dsDNA genome. Herpesviruses are divided into the groups of α, β, and γ herpesviruses. [4]. The latter comprise Kaposi’s sarcoma associated herpesvirus (KSHV) and Epstein-Barr virus (EBV) that are meanwhile recognized carcinogenic herpesvirus types [5].

Squamous cell carcinoma (SCC) is a common malignant tumor disease in humans and animals. Disease arises from keratinocytes, with early intraepithelial lesions (plaques, papillomas, and carcinomas in situ; CIS) ultimately progressing to invasively growing, metastasizing SCCs [6,7,8]. In humans, virtually 100% of cervical cancers, about 50% of genital SCCs, and up to 50% of head-and-neck SCCs (HNSCCs) are caused by infection with high-risk human papillomaviruses (hrHPVs), notably HPV types 16 and 18 [9]. In addition, there is growing evidence that EBV has an active role in the development of hrHPV-unrelated HNSCCs, notably nasopharyngeal carcinomas [10,11].

SCCs are also diagnosed in horses, where they constitute the most common malignant epithelial cancer disease [12]. Equine SCCs can develop anywhere on the skin, yet predominate in the head-and-neck, the ocular, and the genital region [12,13]. Given the invasiveness of equine SCCs, surgical excision is the current therapy of choice. In severe cases, this may necessitate exenteration of the affected eyes or the en-bloc resection of affected external genitalia [6,14,15]. Due to their discrete location, equine HNSCCs are usually discovered at late stages and are not amenable to treatment at this point [16]. Despite the pronounced veterinary impact of disease, the etiology of equine SCC is still incompletely understood. It is widely accepted today that virtually 100% of equine genital SCCs and precursor lesions are (co-)induced by equine papillomavirus type 2 (EcPV2) infection [17,18]. In addition, there is growing evidence that EcPV2 is also associated with a subset of HNSCCs [16,19,20], but etiologically unrelated to equine ocular SCCs [17,18]. Since the discovery of EcPV2 [17], seven additional EcPV types (EcPV3-9) have been identified [21]. There are indications that EcPV3-7 might cause aural plaques [18]. EcPV3 DNA was also found in a subset of genital lesions [22], whilst EcPV8 appears to be associated with cutaneous papillomas [23]. Despite these findings, the respective role of EcPV3-9 in equine mucocutaneous disease remains to be elucidated. There is no information on the prevalence and possible role of equid herpesviruses in SCC so far.

Given the incomplete understanding of equine SCC etiopathogenesis, and recent evidence pointing to EBV promoting human HNSCC development, we screened a series of equine SCCs/precursor lesions and site-matching samples from tumor-free horses for the presence of EcPV2-5 DNA by type-specific PCR. Furthermore, we comparatively tested these samples for the presence of herpesvirus DNA using a consensus herpesvirus PCR approach, followed by amplicon sequencing to identify detected herpesvirus types.

## 2. Materials and Methods

*Ex vivo material*. Sample material subjected to EcPV2-5 and herpesvirus DNA screening was obtained from horses with HNSCCs, (peri-)ocular SCCS, genital SCCs, or respective precursor lesions, and from tumor-free control horses (Table 1; Appendix A).

Histopathological tumor diagnoses were routinely carried out at the Veterinary University’s Institute of Pathology according to established standard procedures. Diagnoses (except for one HN lesion, see horse MIL, Appendix A) were deposited in the University’s Animal Information System (“Tierinformtionssystem”; TIS), from which they were retrieved.

The rationale underlying the choice of control material was that EcPV2 can be detected in this material in horses bearing EcPV2-positive SCCs, related precursor lesions, and occasionally tumor-free individuals [20,24,25,26], and that herpesvirus DNA is commonly found in epithelial swabs and body fluids from healthy horses [5,27,28,29,30,31,32].

Tumor samples and other patient-derived material were obtained between 2008 and 2021 during therapeutic surgical excisions or postmortem; and control samples were collected non-invasively. All samples were taken with the owners’ written consent. 

In many cases, DNA extracted from native sample material was already available (stored at −20). In the other cases, archival (−20 or −80 °C) or fresh native material was subjected to DNA isolation using a DNeasy Blood and Tissue Kit (Qiagen, Hilden, Germany) according to the instructions of the manufacturer. Prior to virus screening, the PCR-compatible quality of all DNA isolates was successfully confirmed by equine β-actin PCR, as previously described [33].

Appendix A provides detailed information on all horses from which samples were collected, including breed, age, coat color, and gender. A short description of the tumor and the diagnosis are provided for the patient group, in addition to the reasons for veterinary consult (Vet consult) for tumor-free control horses. In addition, Appendix A displays the EcPV- and herpesvirus PCR results obtained for each sample (Appendix A).

*Screening for EcPV DNA*. EcPV2, EcPV3 and EcPV5 E6 or E7 PCRs were conducted from 2-µl DNA aliquots exactly as described previously [16]. For many HNSCCs, data on EcPV2, 3, and 5 infections were already available [16]. Screening was repeated for these samples, as well as for a series of previously tested genital lesions, to allow for accurate comparability with respect to EcPV and herpesvirus infection [20,26]. EcPV4 PCR was carried in an analogous manner using primers 5´EcPV4-7143 (5´-CCAAGTTGCTGCAAGATTCTGCACAG-3´) and 3´EcPV4 -461 (3´-GGATCCCTCGTCGTTGCGAACC-3´) for amplification of an 872-bp product comprising the E6 open reading frame.

*Screening for herpesvirus DNA.* A consensus nested PCR was performed from all samples according to VanDevanter et al., with minor modifications, i.e., using Taq DNA polymerase (Roche, Vienna, Austria) according to the manufacturer’s instructions, and adding 9% dimethyl sulfoxide (DMSO, Sigma-Aldrich, Vienna, Austria) to reaction mixtures. This PCR system allows for the detection of most herpesvirus types, including all equid herpesviruses known so far [34]. The first PCR was conducted from 5-µl DNA aliquots, and the nested PCR from 1-µl reaction mixtures. The cycling program for the first and the nested PCR consisted of 45 cycles (92 °C × 30 s; 46 °C × 60 s; 72 °C × 60 s) followed by a final elongation step at 72 °C for 5 min. 

*Amplicon processing*. Amplification products (16 µL) were analysed by 1.5% TAE gel electrophoresis and visualized by ethidium bromide staining. Nested-PCR amplification products of correct size were gel-extracted using a QIAEX II Gel Extraction Kit (Qiagen) and subjected to bi-directional sequencing (Eurofins, Vienna, Austria). Amplicon sequences were identified by BLAST alignment (https://blast.ncbi.nlm.nih.gov/Blast.cgi; repeatedly accessed in 2021 and 2022).

*Statistics.* The significance of respective differences regarding tumor prevalence in relation to (i) gender, (ii) coat color, and (iii) breed was determined by a Chi-square test for goodness-of-fit (https://www.socscistatistics.com/tests/goodnessoffit/default2.aspx; accessed on 20 November 2022. The significance of respective differences regarding (i) EcPV2 and (ii) herpesvirus detection rates in tumor-bearing versus control horses was calculated by a Chi-Square test (https://www.socscistatistics.com/tests/chisquare2/default2.aspx; accessed on 20 November 2022). The statistical significance was uniformly set at *p* < 0.05.

## 3. Results

### 3.1. Tumor Prevalence with Respect to Age, Gender, Coat-Color, and Breed

The average age of tumor patients was the following: 19.9 years in the HN tumor group; 14.7 years in the (peri-)ocular tumor group; and 21.2 years in the genital tumor group. This finding emphasizes that ocular lesions preferentially developed in medium-aged adult horses, whilst HN and genital lesions mainly affected old individuals. HN and ocular lesions were evenly distributed among mares and geldings, whilst castrated males presented significantly more often with genital SCCs or precursor lesions than mares in this study (*p* < 0.05; Table 2). Ocular lesions were significantly more often diagnosed in sorrel-colored horses of Haflinger breed (*p* < 0.05; Table 2), whilst genital tumors predominated in Icelandic horses despite their limited presence in Austria (Table 2).

### 3.2. EcPV DNA Detection Rates in Tumor Patients versus Tumor-Free Horses

EcPV2 DNA was intralesionally detected in 10/22 horses with HNSCCs or CIS affecting the sinonasal, oral and/or pharyngeal regions. Interestingly, an SCC of the maxillary sinus (FIL), and a metastasizing mandibular SCC (SPP) scored positive for EcPV5 DNA (Appendix A). In the HN tumor group, EcPV2 DNA was also detected in salival DNA of a horse (SAM) affected by an EcPV2-positive, pharyngeal metastasizing SCC (SAM), and from a vulvar swab of a mare (OLG) suffering from an EcPV2-negative nasopharyngeal neoplasia (Appendix A). Unfortunately, no follow-up information is available for the latter. Salival and periodontal fluid DNA derived from tumor-free horses scored negative for all EcPV types assessed. When comparing HN tumor-bearing patients with control horses, EcPV2 was significantly more frequently detected in tumor-affected than tumor-free individuals with *p* < 0.0005.

In the (peri-) ocular tumor group, only the two periocular lesions (lesions with a cutaneous portion) included in the study (ALI, MCH) tested EcPV2-positive, whilst all sensu stricto (s. str.) ocular lesions, as well as ocular swabs from control horses, scored negative for this virus type. However, a conjunctival SCC (OCO) was shown to harbor EcPV4 DNA (Appendix A). EcPV2 detection rates did not significantly differ between (peri-)ocular tumor-affected and tumor-free control horses.

As expected on the basis of previous findings, EcPV2 DNA was present in 100% of genital SCCs and precursor lesions (papillomas, CIS), as well as in the smegma of tumor-affected male horses [17,20,26]. In addition, genital swabs from a control horse tested EcPV2-positive (ALE; Appendix A). EcPV3 was not detected in any of the samples from the patient and the control group. Similarly, PBMCs collected from EcPV2-infected tumor patients scored consistently negative. Differences of EcPV2 detection rates in genital tumor-bearing (100%) versus control horses (1.2%) were highly significant with *p* < 0.00001. All results are summarized in Table 3 and presented in detail in Appendix A.

### 3.3. Consensus Herpesvirus PCR Results and Detected Herpesvirus Types

Screening for the presence of herpesvirus DNA was conducted by consensus nested PCR. Following gel electrophoresis and gel extraction, amplification products of expected size and sufficient concentration were subjected to bi-directional sequencing followed by sequence alignment using BLAST to identify detected herpesvirus types. Except for the smegma of a penile SCC patient (ALV), and a penile SCC (MEL) (Appendix A) that tested positive for equine herpesvirus type 3 (EHV3), only γ-herpesviruses, i.e., equine herpesvirus types 2 and 5 (EHV2, EHV5), and asinine herpesvirus 5 (AsHV5) were identified in this study. The obtained results are summarized in Table 4 and shown in detail in Appendix A. Some lesions harbored more than one herpesvirus type, in addition to EcPV DNA (see Section 3.4).

In HN tumors, herpesvirus DNA (EHV2, EHV5, AsHV5) detection rates significantly exceeded those in the control group (63.6 versus 9.6%; *p* < 0.05) (Table 4). In the (peri-) ocular tumor versus the control group, a trend towards ocular lesions harboring herpesvirus DNA more frequently than ocular swabs from tumor-free horses (66.6 versus 46.6%) was noted. Male genital tumor patients and corresponding controls revealed equal herpesvirus infection rates. In mares, however, the presence of herpesvirus infection was generally lower and almost restricted to tumor-free horses with a 24.5% detection rate (Table 4).

### 3.4. Equine Papillomavirus and Herpesvirus Co-Infections

Whilst herpesvirus DNA-positive control samples scored EcPV-negative throughout the study, a proportion of tumors harbored EcPV in addition to herpesviral DNA, as summarized in Table 5:

## 4. Discussion

SCC is the most common malignant cancer disease in horses. Although it seriously impairs the health and welfare of affected individuals, its etiology is incompletely understood, thus hampering the development of more potent diagnostic, prophylactic and therapeutic tools [2,12,16,18,22,35]. Given that information on the exact etiopathogenic role of papillomaviruses in equine SCC is still lacking and that a possible involvement of herpesviruses in this disease has not been addressed so far, we comparatively screened HN, (peri-)ocular, and genital SCCs, or precursor lesions at the respective sites, as well as site-matched samples from tumor-free control horses for the (simultaneous) presence of EcPV type 2-5, and herpesvirus DNA.

Tumor samples stemmed from a total of 26 mares and 53 geldings. While HN and ocular lesions were evenly distributed between genders, geldings were significantly more often affected by genital tumors than mares (36 versus seven patients). There is robust evidence of EcPV2 being causally associated with equine genital SCCs [17,19,20,22,24,26,36,37,38,39]. This is further corroborated by our finding of 100% of genital lesions scoring EcPV2-positive. It has been shown that smegma constitutes a rich reservoir of EcPV2 DNA and putative virus particles in equine penile SCC patients [26]. In stallions, genital hygiene is successfully achieved by the frequent protrusion and retraction of the penis. Similarly, the anatomy and mucosal lining of the mares’ genitalia provide protection from infections. Geldings tend to urinate whilst keeping the penis (partly) retracted, thus promoting the accumulation of smegma, eventually leading to smegmalith formation [6,40]. This observation likely explains the significantly higher incidence of genital SCCs in geldings, as proposed previously [6,26].

Interestingly, 14 of 36 horses presenting with genital SCCs were of Icelandic breed. This finding was somewhat unexpected, since 7240 Icelandic horses were registered in Austria in 2021, corresponding to only 5.5% of the Austrian horse population (https://www.feif.org/feif/members/austria/; accessed on 19 November 2022). Genital SCCs were diagnosed in horses aged 21.2 years on average. The oldest genital SCC patient in this study was a 30-year-old Icelandic gelding. This finding agrees with the concept that genital SCC is a disease of old individuals [6,12,18]. The pronounced longevity of Icelandic horses might explain the relatively high number of Icelandic horses with genital SCCs in this study.

SCCs involving the nictitating membrane or conjunctiva were predominantly diagnosed in horses of the Haflinger breed (13/24). This observation matches the reported association of a missense mutation in damage-specific DNA-binding protein 2 (DDB2) with ocular SCC in Haflinger horses [41,42,43]. Whilst the only two periocular SCCs included in this study scored EcPV2-positive, all ocular lesions as well as ocular swabs from tumor-free horses scored EcPV-negative, thus further corroborating the current theory that ocular SCCs are not associated with PV infection [17,18,19,24,44].

Importantly, 45.5% of HN lesions tested positive for EcPV2. In addition, two further HNSCCs were shown to harbor EcPV5 DNA, whilst saliva and periodontal fluid samples from tumor-free horses scored EcPV2-5-negative throughout the study. This finding points to EcPV2 having an active role in equine HNSCC, as suggested previously [16,20]. In human HNSCC, hrHPV-infection is tightly associated with lesions affecting the oropharynx, including the tonsils and base of the tongue, whilst hrHPV-unrelated tumors have no defined predilection sites in the HN region [45]. In horses, a similar distribution pattern could not be observed. EcPV2-positive HNSCCs were found to involve the sinonasal region, oral cavity, and pharynx at a similar frequency. 

EcPV5 were first identified by Lange et al. [46] from a case of aural plaques and classified as Dyoiotapapillomavirus on the basis of its genetic similarity to EcPV2 [21]. The predicted EcPV5 E6 lacks the PDZ binding domain (XS/TXV/L) that is characteristic of carcinogenic PVs [47]. Hence, a possible association of EcPV5 with HNSCCs appears rather unlikely. This also applies to related Dyoiotapapillomavirus EcPV4 [21] detected from a single ocular lesion.

The consensus nested PCR system established by VanDevanter et al. allows for the detection of all equid herpesviruses known so far [34]. Nonetheless, the use of this system for the screening of equine SCCs and site-matched controls resulted in the almost exclusive detection of γ-herpesviruses EHV2, EHV5, and AsHV5. EHV2 and EHV5 are endemic in all horse populations worldwide [29,48,49,50]. Similarly, AsHV5 has been found in symptomatic and asymptomatic equids [32]. Detection of EHV2, EHV5, and AsHV5 in clinically normal equids and individuals with various disease symptoms has seriously hampered the establishment of virus type-specific etiopathological associations for a long time [28]. Studies on a possible role of these or other herpesvirus types in SCC development and progression have not been conducted so far. Overall, our findings indicate that research should be conducted in this direction. 

In HN tumor patients, herpesvirus DNA was detected in 63.6% of individuals, and identified as EHV2, EHV5, or AsHV5 DNA in 54.5% of cases. Herpesvirus-positive sample material mainly consisted of DNA extracted from tumor tissue. In addition, all nasal swabs and saliva samples collected from HN tumor patients tested positive (6/6). In contrast, only three of 31 saliva and periodontal fluid samples collected from tumor-free horses harbored herpesvirus DNA. These data strongly suggest that research in this area should be pursued.

Herpesviruses have the intrinsic ability to establish latent infections in specific cell subsets to assure their live-long persistence in the infected host. [5]. Cell types and viral proteins supporting herpesvirus latency greatly vary between herpesvirus types [4,5,29,51,52]. In EBV-associated HNSCCs, latent viral gene products including EBV-encoded small RNAs (EBERs), EBV nuclear antigens (EBNAs), and latent membrane proteins (LMPs) are expressed in lymphocytes and epithelial cells, where they promote immune escape and invasive tumor growth [10]. 

In equids, γ-herpesvirus latency and cell tropism are still poorly understood. EHV2 is described to latently reside in B lymphocytes. In addition, tissue macrophages, epithelial Langerhans cells, as well as the ciliary and trigeminal ganglia seem to represent sites of latent EHV2 infection [29,53]. EHV2 has also been identified in, for example, the skin, esophagus, conjunctiva, and vagina, but no information is available as to the type of EHV infection (latent versus lytic) at these sites [29]. EHV5 can establish latent infections in B and T lymphocytes, and possibly alveolar macrophages [27,54,55,56]. Moreover, EHV5 antigens have been detected from alveolar pneumocytes and interstitial fibroblasts of EMPF-affected horses, suggesting that these cell types are permissive for EHV5 infection [50]. Human SCCs were recently identified as containing normal and tumor-specific keratinocytes as well as infiltrates of stromal fibroblasts and different types of immune cells [57]. Consequently, our finding of 63.6% of HN tumors harboring herpesviral DNA currently provides no information on the cell type(s) infected. However, high detection levels in HNSCCs versus low detection levels in oral secretions from tumor-free horses open the intriguing possibility of an active involvement of γ-herpesviruses in equine HNSCC. Future work will focus on the HNSCC cell subsets harboring these viruses and their transcriptional activity in these cells to address this issue.

Equid γ-herpesviruses were also detected from tumor tissue in 66.6% of horses with (peri-)ocular lesions. Based on this finding, in-depth studies are required to determine whether these herpesvirus types are etiopathological agents, additional promotors, or innocent bystanders in (peri-)ocular SCC development. In addition, 46.6% of ocular swabs obtained from tumor free horses scored positive for (γ-)herpesvirus-DNA. The latter finding agrees with the reported prevalence of EHV2 in ocular secretions of apparently healthy horses and the assumed tropism of the virus for conjunctival Langerhans cells [53]. Similarly, AsHV5 DNA was detected from nasal and conjunctival swabs of clinically normal Lipizzaner horses in Austria, suggesting that subclinical AsHV5 infection may constitute a more frequent event in equine populations than previously reported [58]. 

Equine γ-herpesvirus DNA was detected in penile SCCs and genital swabs from tumor-free male horses at a similar ratio (47.2 versus 47%). In mares, γ-herpesvirus DNA was detected to a lesser extent, and preferentially in genital swabs from healthy individuals (24.5%). This finding points to genital hygiene having a possible role in γ-herpesvirus infection. Similar to what was observed for EcPV2, smegma/cellular debris accumulating with vaginal secretions may act as a reservoir for γ-herpesviruses. This assumption is also supported by the isolation of EHV2 from the equine genital tract [59].

As described, EBV, EHV2 and EHV5 possess gene homologues to host cell genes coding for specific cytokines and their receptors. One of these gene homologues, E7, encodes an interleukin-10 (IL-10)-like protein resembling the BCRF1 protein expressed by EBV [60,61]. In EBV infections, viral IL-10 (vIL-10; BCRF1) acts as an anti-inflammatory cytokine, reduces the cytotoxic activity of natural killer (NK) cells, and impairs adaptive immunity [62,63,64]. Recent studies also point to vIL-10 negatively impacting the phagocytic activity of monocytes, thus reducing antigen presentation, and by this, the immune recognition of EBV infection [62,63]. Given the high structural similarity of EBV and EHV2/EHV5 vIL-10 [29], analogous functions may be attributable to vIL-10 in EHV2 und EHV5 infection. As observed for KSHV, the open reading frame 74 of EHV2 and EHV5 codes for a homologue of an IL-8 receptor that is commonly displayed on macrophages and endothelial cells and that is involved in neutrophil chemotaxis [61]. Previous findings suggest that this receptor homologue may be involved in host immune response modulation by the downregulation of monocyte chemoattractant protein I (MCP-1; CCL2) [65], which is required for immune cell recruitment to sites of viral infection [66].

The mechanisms underlying immune evasion by equid γ-herpesviruses are still poorly understood. However, current knowledge on EBV- and KSVH-mediated immune modulation and first insights into EHV2 and EHV5 infection and latency support the theory that equid γ-herpesviruses may at least indirectly contribute to equine SCC pathogenesis by negatively interfering with antigen recognition pathways, promoting the survival and proliferation of neoplastic cells, and possibly amplifying the carcinogenic activity of EcPV2 in cases of coinfection.

EHV2 and EHV5 also code for latency associated nuclear antigen (LANA), a homologue to EBNA1 [55]. The latter protein is expressed throughout latent EBV infection and is crucial for viral episome maintenance and replication in synchrony with cell division [67]. In vitro, EBNA1 has been shown to inhibit the functional activation of the immune regulatory protein complex NF-κB in carcinoma cells [68]. Furthermore, EBNA1 activates the transcription factor AP-1, leading to the overexpression of pro-tumoral c-Jun whilst inhibiting TGF-β1 expression [62]. In addition to promoting tumor growth, EBNA1 also contributes to the creation of a protumoral microenvironment in nasopharyngeal carcinoma by the chemoattraction of regulatory T cells [69]. These findings highlight the carcinogenic role of EBNA1 and suggest that its homologue LANA may confer equid γ-herpesviruses with similar oncogenic properties.

This work further strengthens the concept of EcPV2 acting as an oncovirus and is the first to provide evidence of the presence of equid γ-herpesvirus DNA in equine SCCs affecting the HN, ocular, and genital region of horses. Further investigations on a possible direct or indirect role of these herpesviruses in equine SCC onset and progression remain challenging, inter alia, by their omnipresence in equid populations. However, research in this area may be highly rewarding given the veterinary relevance of SCC and the incomplete understanding of the etiopathological significance of EHV2, EHV5 and AsHV5.

## Figures and Tables

**Table 1 pathogens-12-00179-t001:** Number and type of samples from tumor patients and control horses.

Sample Source	Number of		Type of Samples	Number of Samples per Type
	Horses	Samples		
Group 1: Horses with HNSCCs (77.2%) or precursor lesions	22	33	Tumor tissue	21
			Nasal swabs	3
			Lymph node tissue	2
			Saliva	3
			Tissue from ocular lesion	1
			Genital swabs	1
			Intact mucosa	1
			PBMCs	1
Tumor-free control horses for Group 1	31	31	Saliva	26
			Periodontal fluid	5
Group 2: Horses with (peri-)ocular SCCs (79.1%) or precursor lesions	24	36	Ocular tumor tissue	23
			Periocular tumor tissue	2
			Tissue from metastases	2
			Intact conjunctiva/nictitating membrane	3
			Oral SCC	1
			Ocular swabs	1
			Ocular tumor swab	1
			PBMCs	3
Tumor-free control horses for Group 2	30	30	Ocular swabs	30
Group 3: Horses with penile SCCs (83.3%) or precursor lesions	36	53	Tumor tissue	34
			Tumor swabs	3
			Tumor cytobrush	1
			Smegma	7
			Ocular swabs	3
			Intact skin	1
			Primary tumor cells	2
			PBMCs	2
Tumor-free control horses for Group 3	34	34	Swabs from urethral fossa	9
			Smegma	25
Group 4: Horses with vulvar SCCs (85.7%) or precursor lesions	7	7	Vulvar tumor tissue	6
			Tumor cytobrush	1
Tumor-free control horses for Group 4	49	49	Vulvovaginal swabs	49

PBMCs: peripheral blood mononuclear cells.

**Table 2 pathogens-12-00179-t002:** Tumor prevalence in relation to gender, coat-color, and breed.

Parameter	Description	HN Tumors (22 Patients)	(Peri-)Ocular Tumors (24 Patients)	Genital Tumors (43 Patients)
Gender	Mares	8	11	7
	Geldings	14	13	36 *
Coat color	Black	4	0	4
	Bay	3	2	8
	Chestnut	7	2	9
	Sorrel	4	13 *	6
	Grey	2	3	4
	Other *	2	4	11
Breed	WB	4	5	10
	Trotter	3	0	0
	Haflinger	4	13 *	5
	Icelandic horse	3	0	14
	Pony	4	2	4
	Other ^§^	4	5	10

* *p* < 0.05; ^§^ Including horses of unknown color/breed.

**Table 3 pathogens-12-00179-t003:** EcPV2-5 DNA detection rates in tumor patients versus control horses.

	EcPV2	EcPV3	EcPV4	EcPV5
HN tumor patients	10/22 (45.5%) *	0/22	0/22	2/22 (9%)
Control horses	0/31	0/31	0/31	0/31
(Peri-)ocular tumor patients	2/24 ^§^ (8.3%)	0/24	1/24 (4.1%)	0/24
Control horses	0/30	0/30	0/30	0/30
Male genital tumor patients	36/36 (100%) *	0/36	0/36	0/36
Control horses	1/34 (2.9%)	0/34	0/34	0/34
Female genital tumor patients	7/7 (100%)	0/7	0/7	0/7
Control horses	0/49	0/49	0/49	0/49

* *p* < 0.05; ^§^ these were the only two periocular SCCs in this study. EcPV2-EcPV5: equine papillomavirus types 2, 3, 4, and 5.

**Table 4 pathogens-12-00179-t004:** Herpesvirus detection rates in tumor patients versus control horses.

Horses	Herpesvirus-Positive (in %)	EHV2	EHV5	AsHV5	Type NI
HN tumor patients	14/22 (63.6%)	5/22	4/22	5/22	2/22
Control horses	3/31 (9.6%)	1/31	0/31	1/31	1/32
(Peri-)ocular tumor patients	16/24 (66.6%)	5/24	5/24	2/24	5/24
Control horses	14/30 (46.6%)	4/30	3/30	2/30	5/30
Male genital tumor patients	17/36 ^§^ (47.2%)	6/36	4/36	4/36	1/36
Control horses	16/34 (47.0%)	9/34	2/34	4/34	1/34
Female genital tumor patients	1/7 (14.2%)	0/7	0/7	1/7	-
Control horses	12/49 (24.5%)	5/49	1/49	7/49	-

^§^ Two samples scored positive for EHV3; NI: not identified (too low amplicon yields). EHV2: equine herpesvirus type 2; EHV5: equine herpesvirus type 5; AsHV5: asinine herpesvirus type 5.

**Table 5 pathogens-12-00179-t005:** Numbers of horses with intralesional herpesvirus/EcPV co-infections.

Horses	Herpesvirus-Positive	EcPV2	EcPV4	EcPV5
HN tumor patients	14/22	6/14	0/14	1/14
(Peri-)ocular tumor patients	16/24	2/16	1/16	0/16
Male genital tumor patients	17/36	17/17	0/17	0/17
Female genital tumor patients	1/7	1/1	0/1	0/1

EcPV2-EcPV5: equine papillomavirus types 2, 3, 4, and 5.

## Data Availability

All data are provided in the article and Appendix A.

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
