# Peer review of "Detection of Equine Papillomaviruses and Gamma-Herpesviruses in Equine Squamous Cell Carcinoma"

_pathogens, 2023, doi:10.3390/pathogens12020179_

Round 1
Reviewer 1 Report
attached

Reviewer 2 Report
The paper Detection of equine papillomaviruses and gamma-herpesviruses in equine squamous cell carcinoma is a well written and interesting paper on biomolecular detection of viral agents from tumours. I would like to submit you some observations:
The first issue is comparing tumour data against “other samples” data. I understand the reason for which you did not perform biopsy, but please add a paragraph and reference to support the appropriateness of the approach.
Why did you not compare statistically the same samples collected from controls and cases?
Please add a justification for which you choose a p of 0.1 and not of 0.05.
The headings of the tables must be revised explaining the bold and all the virus names, remind that the tables and figure should be comprehensible also standing by themselves.
The discussion in general is good, but in some parts it seems more a review, and it is not always connected with the results. Please revise.
Round 2
Reviewer 1 Report
I think that the revised version of this work is now ready for publication in the journal
Reviewer 2 Report
Dear authors,
thank you for taking into account all the comments and revise the text accordingly. I think that now the paper is acceptable for publication.
I prefer not to send it back with minor revisions, but I strongly advice to add in the tables reporting the results a different symbol for the notes and use the asterisks for indicate the statistical differences, adding at the end of the table *= p<0.05. This little change made, the paper would be ready.